# Early Detection Is the Best Prevention—Characterization of Oxidative Stress in Diabetes Mellitus and Its Consequences on the Cardiovascular System

**DOI:** 10.3390/cells12040583

**Published:** 2023-02-11

**Authors:** Sanela Rajlic, Hendrik Treede, Thomas Münzel, Andreas Daiber, Georg Daniel Duerr

**Affiliations:** 1Department of Cardiothoracic and Vascular Surgery, University of Medicine Mainz, 55131 Mainz, Germany; 2Center for Cardiology, Department of Cardiology, Molecular Cardiology, University Medical Center, 55131 Mainz, Germany; 3German Center for Cardiovascular Research (DZHK), Partner Site Rhine-Main, 55131 Mainz, Germany

**Keywords:** oxidative stress, reactive oxygen species, diabetes

## Abstract

Previous studies demonstrated an important role of oxidative stress in the pathogenesis of cardiovascular disease (CVD) in diabetic patients due to hyperglycemia. CVD remains the leading cause of premature death in the western world. Therefore, diabetes mellitus-associated oxidative stress and subsequent inflammation should be recognized at the earliest possible stage to start with the appropriate treatment before the onset of the cardiovascular sequelae such as arterial hypertension or coronary artery disease (CAD). The pathophysiology comprises increased reactive oxygen and nitrogen species (RONS) production by enzymatic and non-enzymatic sources, e.g., mitochondria, an uncoupled nitric oxide synthase, xanthine oxidase, and the nicotinamide adenine dinucleotide phosphate (NADPH) oxidase (NOX). Considering that RONS originate from different cellular mechanisms in separate cellular compartments, adequate, sensitive, and compartment-specific methods for their quantification are crucial for early detection. In this review, we provide an overview of these methods with important information for early, appropriate, and effective treatment of these patients and their cardiovascular sequelae.

## 1. Diabetes Mellitus

Diabetes mellitus (DM) comprises metabolic disorders sharing the hallmark of continuous hyperglycemia [1]. It is caused either by pancreas dysfunction producing an insufficient amount of insulin (a hormone that regulates blood sugar or glucose; type 1 DM) or when insulin-sensitive tissues do not respond appropriately to insulin (type 2 DM) [2]. Additionally, during pregnancy, gestational DM may develop and cause complications at birth, increasing the mother’s risk of type 2 DM (T2DM) and obesity of the offspring [3].

DM-associated vascular damage results in high morbidity and mortality in our society, and around 50% of people suffering from DM die due to cardiovascular complications [4,5,6]. This metabolic imbalance leads to severe cellular/organ damage, primarily affecting the blood vessels and nerve tissue [7]. The main reason for the development of T2DM in our society is the increasing economic wealth accompanied by easy accession to high-calorie food in western countries [3]. Accordingly, the WHO documented an increase in the incidence of DM in adults from 180 million in 1980 to 422 million in 2014. The global prevalence (age-standardized) of DM has nearly doubled since 1980, rising from 4.7% to 8.5% in the adult population. Age-standardized prevalence trends are shown in Figure 1 [2].

In the 10th edition of “Diabetes Atlas”, the “International Diabetes Federation” (IDF) estimated approximately 6.7 million deaths (age 20–79), 1 every 5 s, as a result of DM and its complications in 2021 [8]. The total number of deaths combines the number of annual all-cause deaths due to DM-related disease in each geographic region stratified by age and sex as well as age- and sex-specific mortality relative risks in people with and without DM. IDF’s method for estimating DM-related mortality is described in more detail elsewhere [9,10,11]. The IDF region with the highest estimated number of DM-related deaths in 20–79 years old is West Pacific, with approximately 2.3 million deaths. This is followed by EUR, with about 1.1 million deaths, as shown in Figure 2 [8].

DM, along with its consequences, is causing an enormous economic loss to affected subjects, their families, the health systems, and national economies through direct medical costs and loss of work and wages. Hospital and outpatient care are costly, but a contributing factor is also a rise in costs for insulin analogs, which are increasingly prescribed despite little evidence of significant advantages compared to cheaper human insulins [2].

Very often, T2DM stays undetected for quite a long time and gets discovered accidentally through a blood or urine glucose test. Therefore, it is often diagnosed in the more advanced stages when it is already coupled with significant cardiovascular complications. Based on this scenario, a better understanding of the mechanisms underlying diabetic vascular disease (Figure 3) is necessary because it may help provide more causal treatment approaches in order to prevent or delay the development of its devastating personal and socioeconomic consequences [12].

The current diagnosis of prediabetes and diabetes is based on the blood glucose or hemoglobin A1C (HbA1C) test. However, both methods are invasive and time-consuming, with high costs. Moreover, methylglyoxal (MG) is evolving as a DM marker because it plays a vital role in biological processes [13]. In addition to glucose in human blood, there are also considerable amounts in other biofluids such as saliva, tears, sweat, and interstitial fluids (ISF). However, scientists showed that ISF reflects the change in blood glucose levels with an unavoidable delay [14,15]. Furthermore, since it was shown that early damage of small nerves could occur during the progression of T2DM, some devices have emerged to assess small-fiber autonomic dysfunction [16,17]. On top of the mentioned approaches, others are mentioned in the literature, but none have found a place in clinical use. Therefore, the focus of this review is not on therapy and the diagnosis of DM but, instead, on the early detection of associated detrimental oxidative stress that consequently contributes to DM-induced CVD. Of note, a large part of diabetic pathophysiology is related to cardiovascular sequelae, which without any doubt contribute to oxidative stress and accordingly show overlap with the redox markers for diabetes itself. However, we think that the initial formation of reactive oxygen and nitrogen species comes from hyperglycemia-related processes that cause further damage to the islet cells, thereby aggravating the diabetic condition with a further increase in oxidative stress. Only later, secondary cardiovascular damage by the development of hypertension and progression of atherosclerosis will contribute to oxidative stress conditions, making it impossible to distinguish between the different causalities. Therefore, a combination of the measurement of oxidative stress markers or measurement of ROS (reactive oxygen species) and harmful RNS (reactive nitrogen species) with classical markers of hyperglycemia-driven pro-oxidative pathways (e.g., advanced glycation end products (AGE) and their receptor (RAGE)) may help to more specifically identify and correctly assign oxidative stress markers for early diabetes.

## 2. Oxidative Stress

A significant cause of DM is oxidative stress. Under normal physiological conditions, the production of ROS and harmful RNS is low, and the antioxidant system quickly removes both before they induce any structural and functional damage. However, this balance is fragile; accordingly, ROS- and RNS-induced cellular damage appears frequently, and the damaged molecules must be repaired or replaced, as described in vivo in animals and humans [18].

Sies et al. were the first to introduce the term “oxidative stress” to describe the imbalance between the pro-oxidants and antioxidants in favor of the oxidative processes [19]. Since then, redox biology has developed into various disciplines, such as chemistry and radiation biology, via biochemistry and cell physiology to general biology and medicine [20]. An early concept was that the principal processes of redox regulation, named “redox signaling” and “redox control”, promoted the redox reactions in living cells, as previously summarized [21]. Later on, the term “redox biology” gained popularity [22,23]. Furthermore, the concept of oxidative stress is related to “redox signaling,” although the latter term is more connected with the beneficial effects of ROS and RNS [24].

These studies promoted the original idea that oxidative stress in an open metabolic system arises when RONS production overwhelms the intrinsic antioxidant defense capacity; under physiological conditions, a constant balance is kept, thus providing redox balance [25]. Each deviation from redox balance is considered a stress condition that initiates consecutive responses. Oxidative stress can be classified by its intensity to physiological oxidative stress (“eustress”) and pathophysiological oxidative stress (“distress”) (Figure 4) [26].

The central “master regulators” of redox homeostasis are the Nrf2/Keap1 and NF-kB/IkB systems [27,28]. Due to many different pro-oxidants and antioxidants, scientists attempted to classify different categories of oxidative stress and conceptually introduce stress by its intensity, ranging from “eustress” to “distress” [29,30,31]. In addition, acute, chronic, and repetitive oxidative stress has been propagated [32]. As a result, redox homeostasis started to be considered “the golden mean of healthy living” [33].

Increased blood glucose levels in DM cause increased production of ROS by the mitochondria electron transport chain [34]. Furthermore, ROS’s high reactivity further induces chemical changes in virtually all cellular components, leading to DNA and protein modification and lipid peroxidation, further being responsible for the development of CVD [35,36]. Therefore, there is considerable demand in science and medicine for reliable and effective methods in quantitative assessment of the oxidative stress in biological samples, consecutively for the ROS/RNS and antioxidants detection, since the imbalance of these two groups of molecules in favor of the pro-oxidants causes oxidative stress. Therefore, developing new ROS/RNS or antioxidant detection methods is very important for research in the biomedical and pharmacological fields, with an emphasis on DM. Furthermore, it is also essential to assess these species in different cellular compartments and shed light on the resulting pathomechanisms that contribute towards the development and progression of DM-associated CVD.

## 3. Antioxidants

Antioxidants stand on the other side of the redox balance, with opposing effects on the ROS and RNS. Antioxidants can be defined as species that reduce oxidative stress by degrading ROS/RNS to less reactive species or suppressing radical chain reactions within proteins, lipids, carbohydrates, or DNA [37]. A large variety of endogenous and exogenous antioxidants is active in an organism. They can be categorized as enzymatic (endogenous) and non-enzymatic (partly delivered by food intake) [38], as shown in Figure 5 [39]. The efficiency of the antioxidant system is directly connected with the nutritional intake and endogenous antioxidant enzyme (e.g., superoxide dismutase; SOD) or the low molecular weight molecule (e.g., glutathione; GSH or bilirubin) production, which can be improved by exercise training, dietary enrichment or intermittent fasting [37].

## 4. Sources of Oxidative Stress in Diabetes Mellitus

Many studies provide certain pieces of evidence for oxidative stress involvement in DM [40,41,42,43]. Accordingly, both enzymatic and non-enzymatic pathways can generate high glucose-induced ROS, as summarized in Figure 6. The enzymatic pathways include NADPH oxidase, NOX, uncoupling of nitric oxide synthase (NOS), cytochrome P-450 (CYTP450), cyclooxygenase (COX), lipoxygenase (LOX), xanthine oxidase (XO), and myeloperoxidase (MPO). Opposite, the non-enzymatic pathways include mitochondrial electron transport chain (mETC) deficiencies, advanced glycation end products (AGEs), glucose autoxidation, transition-metal catalyzed “Fenton reactions”, and polyol (sorbitol) pathways.

Non-enzymatic oxidative stress sources result from the biochemical oxidation of glucose. Thus, hyperglycemia can directly cause increased ROS generation. For instance, glucose autoxidation generates highly reactive hydroxyl radicals (^∙^OH) [44,45]. Additionally, glucose reacts with proteins in a non-enzymatic manner, leading to the development of “Amadori products,” further forming AGEs. ROS are formed at multiple steps during this process. Additionally, in hyperglycemia, there is an increased metabolism of glucose through the polyol (sorbitol) pathway, which also results in enhanced production of O_2_^∙−^ [36].

Further, the mitochondrial respiratory chain is a non-enzymatic source that generates reactive species. When glucose levels are high, mitochondrial ROS production is enhanced, thus inducing oxidative stress, lipid peroxidation, and tissue impairment [46]. The oxidative phosphorylation process includes electrons transferred from electron carriers NADH and reduced flavin adenine dinucleotide (FADH_2_), through four complexes located on the inner mitochondrial membrane, to oxygen, thereby producing ATP (adenosine triphosphate) [47]. Hyperglycemia and hyperlipidemia enhance glucose and lipid catabolism, leading to enhanced formations of NADH and FADH_2_. This overproduction of NADH can cause higher proton gradient production in mitochondria, with the surplus electrons being transferred to O_2_ to produce toxic O_2_^∙−^ [48].

Nishikawa and colleagues showed that the generation of excess pyruvate via accelerated glycolysis under hyperglycemic conditions floods the mitochondria and causes O_2_^∙−^ generation at Complex II in the respiratory chain. Notably, that blocks O_2_^−^ radicals through three different approaches: either a small molecule uncoupler of mitochondrial oxidative phosphorylation (CCCP), overexpression of uncoupling protein-1 (UCP1), or overexpression of Mn-SOD is used. Further, it prevents changes in nuclear factor κB (NF-κB), polyol pathway, AGE formation, and PKC activity [49].

Moreover, all forms of enzymatic NOS sources require five cofactors/prosthetic groups: flavin adenine dinucleotide (FAD), flavin mononucleotide (FMN), haem, BH_4_, and Ca^2+^-calmodulin. If NOS is missing its substrate L-arginine or one of its cofactors, NOS may produce O_2_^−^ instead of∙NO, and in this state, it is preferred that NOS is uncoupled.

The phagocytic NOX (NOX-2) was the first example of a system with the sole enzymatic function to produce ROS in contrast to the previously identified sources that have ROS as side products. In the subsequent years, scientists screened for proteins in phagocytes that could be responsible for ROS production [50]. They identified a suitable candidate by cloning the gp91phox (NOX2) protein, a catalytic subunit of the phagocyte NOX [51]. NOX-2 is a nonmitochondrial ROS source and plays an essential role in the O_2_^∙−^production through oxygen reduction using the electron donor NADPH as a cofactor. The mammalian NOX protein family comprises seven isoforms: NOX1-5, Dual oxidase 1 (DUOX1), and DUOX2 [52]. Most of them produce O_2_^−^ except NOX4, DUOX1, and DUOX2, which release H_2_O_2_, although it is unclear whether the initial enzymatic product is O_2_^∙−^ that is dismutated within the enzyme’s active site [53,54]. Zhang and coworkers observed that glycated BSA (Gly-BSA) promotes ROS production and increases NOX2 activity in cardiomyocytes [55]. They also reported that the activation of NOX2 was protein kinase (PKC)-dependent and associated with the translocation of the NF-κB to the nucleus. Surprisingly, neither NOX4, xanthine oxidase (XO), nitric oxide synthase (NOS), nor mitochondrial ROS seemed to play a role in this process. Scientists also showed persuasive evidence that PKC, stimulated in DM via multiple mechanisms, i.e., polyol pathway and angiotensin II (AT-II), activates NOX [56]. Therefore, this pathway is also quite important when looking into the pathology of DM.

Furthermore, it was shown that AGE/RAGE signaling also triggers vascular complications via NOX-induced oxidative stress in diabetic rats, mitochondrial ROS production, and inflammation with atherosclerosis [57,58,59]. Macrophages from gp91phox (NOX2) null mice showed a lower response to AGE stimulation. Contrary in cultured endothelial cells, inflammation was activated by AGE envisaged by the upregulation of vascular cell adhesion molecule 1 (VCAM-1) [57]. Scientists showed a correlation between oxidative stress, AGE/RAGE signaling, inflammation, and endothelial function in a model of T2DM (ZDF rats) with empagliflozin (SGLT2 inhibitor) therapy [60], pointing towards vital crosstalk between these parameters [61,62].

## 5. Molecular Mechanisms Leading to Diabetes Mellitus

Previous studies demonstrated an important role of oxidative stress in the pathogenesis of CVD in patients with DM and that hyperglycemia is the origin of elevated oxidative stress [63,64,65,66,67]. In addition, oxidative stress affects β-cells through different mechanisms Figure 7 [68,69]. A healthy and functional mass of pancreatic β-cells is necessary for normal glucose homeostasis, and varying levels of β-cell dysfunction accompany DM. For instance, oxidative stress can initiate apoptotic processes in the pancreatic cells, consequently causing the loss of β-cells [68,69]. Likewise, enhanced free radical species production negatively affects metabolic pathways in the β-cells and impairs K_ATP_ channels leading to lower insulin secretion.

Furthermore, when increasingly produced, radicals can obstruct the nuclear transcription factors involved in insulin gene expression, e.g., insulin promoter factor 1 (Pdx-1) and a transcription factor (MafA), therefore suppressing insulin production at the DNA level [70]. Although it is often assumed that accelerated apoptosis of β-cells is the only underlying pathomechanism leading to T2DM, limitations in β-cell replication and/or neogenesis could be just as important [71]. For example, free radicals play a physiological role in β-cell proliferation, but an excess of free radicals will disturb the β-cell neogenesis [72].

Additionally, oxidative stress affects molecular mechanisms leading to DM on the mitochondrial level since it is often a key player in mitochondrial dysfunction [73]. It can impair mitochondrial function by altering the normal function of the mitochondrial respiratory chains, reducing the respiratory capacity of mitochondria, increasing the proton leak in mitochondrial respiratory chains, altering the potential difference across the inner mitochondrial membrane, and reducing the integrity of the mitochondrial membrane [74,75]. Free radicals can also activate TLRs (toll-like receptors) that, in turn, impair β-cell function. The family of these receptors plays an essential role in alerting the innate immune system of danger, and it seems to be activated by damage-associated molecular pattern molecules (DAMPs) that are released in conditions of oxidative stress [76]. Furthermore, free radicals influence biochemical pathways, including the stress-activated signaling pathways of nuclear factor-κB (NF-κB), NH_2_-terminal Jun kinases/stress-activated protein kinases (JNK/SAPK), and p38 mitogen-activated protein (MAP) kinase, playing a central role in β-cells dysfunction [77].

**Figure 7 cells-12-00583-f007:**
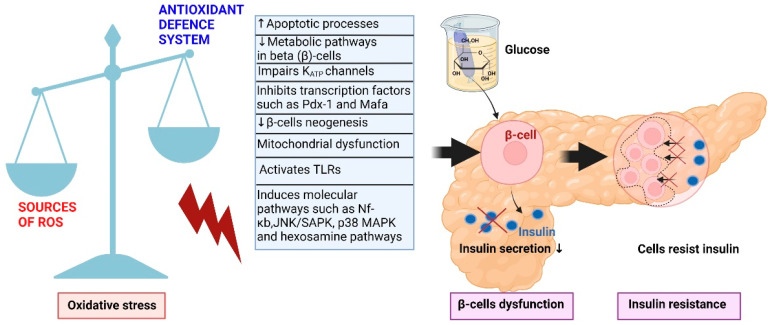
Graphical presentation of possible molecular mechanisms leading to DM. DM, diabetes mellitus; Pdx-1, insulin promoter factor 1; MafA, a transcription factor; TLRs, toll-like receptors; Nf-*κ*b, nuclear factor kappa b; p38 MAPK, p38 mitogen-activated protein kinases; JNK/SAPK, stress-activated protein kinase/c-Jun NH(2)-terminal kinase. The figure is drawn with Biorender, based on data from [75] under the open access Creative Commons license.

## 6. Cardiovascular Disease in Diabetes Mellitus

Since it was shown that myocardial infarction and stroke are the dominant cause of death in patients suffering from DM, it is crucial to understand the adverse effects of hyperglycemia on the vasculature. One of the mechanisms mediating those adverse effects is the activation of protein kinase C (PKC) by hyperglycemia-induced elevated diacylglycerol (DAG) levels, partly due to new synthesis [78,79]. This observation was supported by in vitro studies which showed changes in PKC-specific activities as well as in its isoforms and DAG levels in diabetic rats [80]. However, the essential mechanisms of PKC-mediated endothelial dysfunction are still insufficiently understood. In vitro studies showed that PKC-mediated phosphorylation of nitric oxide synthase (NOS) III protein at Thr495 may reduce the enzyme’s activity [81].

Furthermore, endothelial cells showed increased NOS III expression when stimulated with phorbol esters or glucose [82,83]. Glucose also increases endothelial superoxide production, further elevating peroxynitrite’s vascular formation, a product of nitric oxide (NO)/superoxide reaction [83,84]. It has been shown that peroxynitrite also oxidizes tetrahydrobiopterin (BH_4_), a NOS III cofactor, to dihydrobiopterin (BH_2_) [85]. When BH_4_ is deficient, NOS III is in an uncoupled state, and electron shuttling from the NOS III reductase domain to the oxygenase domain is redirected to molecular oxygen rather than to L-arginine [86,87]. This consequently results in the production of superoxide rather than NO. In this regard, several studies showed that administering the NOS III cofactor BH_4_ improves endothelial dysfunction in DM [88].

Moreover, scientists showed that NOS III expression was not decreased but rather increased by 3-fold in diabetic vessels. In contrast, either the bioavailability or production of∙ NO in these vessels was markedly reduced. Accordingly, it was shown that vascular superoxide was elevated in diabetic vessels and that an extraordinary portion of this seemed to be derived from NOS III itself (in an uncoupled state) and NOX [89].

Over time, hyperglycemia can affect blood vessels, consequently causing different cardiovascular alterations. As previously mentioned, the most common cause of death among patients with DM is CVD [90]. Worldwide, CVD affects approximately 32.2% of all persons with T2DM [91]. On top of the indispensable increase in mortality in diabetic patients, a combination of DM with manifestations of CVD, such as myocardial infarction or stroke, nearly doubles the mortality rate, reducing life expectancy by ≈12 years [92].

In 2019, the “Swedish National Diabetes Register” conducted a study that included 271,174 patients with T2D and compared them with 1,355,870 controls based on age, sex, and county. They were followed for 5.7 years. Patients with five specific risk factors for CVD were included: elevated glycated hemoglobin level, elevated LDL (low-density lipoprotein) cholesterol, albuminuria, nicotine abuse, and elevated blood pressure. The main observation was that T2D patients with specific cardiovascular risk factors did not show a significantly increased risk of death, myocardial infarction, or stroke compared to the control population. However, these subjects had a higher risk of hospitalization for heart failure than control subjects. Furthermore, expanding glycated hemoglobin outside the target range was the primary predictor of stroke and acute myocardial infarction [93]. In 2014, a new possible link between the PPAR-α system, inflammatory reaction, and myocardial remodeling in a diabetic heart was suggested [94].

### 6.1. Type 1 Diabetes Mellitus (T1DM)

When insulin secretion is deficient due to the autoimmune destruction of β-pancreatic cells, it leads to the metabolic disturbances associated with insulin-dependent DM (IDDM) [95]. Global incidence and prevalence of T1DM were estimated to be 234,710 and 9,004,610, respectively, in 2017. High-income countries, with 17% of the global population, accounted for 49% of global incident cases and 52% of prevalent cases [96]. In the recent two decades, the mortality of T1DM has decreased worldwide: a study revealed the decreasing mortality trend of T1DM by different countries, age and sex, and relation to general population mortality between 2000 and 2016 [97].

Disease development represents the end stage of β-cell destruction leading to T1DM. T1DM has been characterized as an autoimmune disease, and Al Homsi and Lukic explained it as follows [98]:Presence of immuno-cells in infiltrated pancreatic holm;Association of susceptibility to disease with the class II (immune response) genes of the major histocompatibility complex (MHC; human leucocyte antigens HLA);Presence of holm cell-specific autoantibodies;Modification of T cell-mediated immunoregulation, in particular in the CD4+T cell compartment;Monokines (e.g., TNF-α and IFN-α/β) and TH1 cells can be involved in producing interleukins in the disease process;Reaction to immunotherapy;Some organ-specific autoimmune diseases can occur in affected individuals or their family members.

### 6.2. Type 2 Diabetes Mellitus (T2DM)

T2DM is a significant risk factor for CVD, the most common cause of death among patients with T2DM. According to the need for insulin therapy, IDDM can be differentiated from non-insulin-dependent DM (NIDDM). Compromised mechanisms such as activated adipose tissue macrophages that are strongly linked to metabolic dysfunction [99], impaired generation of insulin’s second messenger, diminished glucose transport into the cell, or a post-glucose transport defect in some critical enzyme involved in glucose utilization [100], lead to impaired insulin secretion by the pancreatic β-cells and impaired insulin action through insulin resistance.

An estimated 90–95% of adults living with DM worldwide are cases of T2DM [101]. This is probably because T2DM is associated with several other metabolic abnormalities, such as central obesity, hypertension, and dyslipidemia, which contribute to the very high rate of cardiovascular morbidity and mortality [102].

T2DM has long been categorized as a complex metabolic syndrome affected by several factors, such as the irregular metabolism of carbohydrates, fats, and proteins, leading to increased glucose levels and lipids in the blood [103]. Furthermore, chronic exposure to elevated levels of glucose and lipids triggers various pathways responsible for inducing impaired insulin secretion from the β-cells of pancreatic islets, insulin resistance in peripheral tissues, decreased glucose utilization in peripheral tissues, and abnormal hepatic glucose production [104].

Several studies about T2DM pathology showed that ROS and oxidative stress are among the crucial ones responsible for the pathogenesis of insulin resistance, impaired insulin secretion, glucose utilization, abnormal hepatic glucose production, and ultimately overt T2DM by activation of various pro-inflammatory mediators, transcriptional mediated molecular and metabolic pathway including hexosamine pathways, advanced glycation end-products (AGEs) formation, and protein kinase C beta type (PKCβ1/2) [103,104].

## 7. Methods to Access Oxidative Stress

Numerous studies show that oxidative stress is significantly responsible for diabetic complications [105,106,107]. In hyperglycemia-sensitive cells, such as endothelial cells, the increasing levels of glucose cause ROS formation in mitochondria, consequently impairing mitochondrial function and, thus, an endothelial function of large conductance and small resistance vessels [105]. In addition, activated endothelial cells attract monocytes, further enhancing inflammation and promoting macrovascular and microvascular injury. ROS also plays a role in epigenetic modifications involved in the maintenance of pro-inflammatory response [108]. Diabetic conditions also provoke ROS formation in macrophages [109]. Furthermore, the DM-induced secretion of monocyte chemotactic protein CCL2 (MCP1) by endothelium attracts monocytes, whereas the upregulated endothelial surface expression of adhesion molecule VCAM promotes their adhesion and diapedesis [109,110]. Therefore, correctly accessing oxidative stress is of high importance in this field.

The oxidative stress can be accessed through:(i)Testing the effects of acute vitamin C challenges on endothelial dysfunction [111];(ii)Measurements of ROS/RNS directly;(iii)Measurements of oxidative damage markers (stable footprints) or expression levels of the ROS-producing enzymes;(iv)Measurements of levels of components of the antioxidant defense system [112].

Indeed, many studies suggest approaches and methodologies for accessing oxidative stress [113,114].

A detailed classification of the methods for determining and quantifying oxidative stress is presented in Figure 8.

### 7.1. Testing the Effects of Acute Vitamin C Challenges on Endothelial Dysfunction

Increased oxidative stress has been linked to impaired endothelial function in different CVDs and plays an essential role in the pathogenesis of cardiovascular events. Therefore, accessing oxidative stress through testing endothelial function is extremely important and is often used in clinics (e.g., by the acetylcholine test in coronaries in the catheter laboratory or by forearm plethysmography). Moreover, testing the effects of acute vitamin C challenges on endothelial dysfunction is a two-in-one approach since we can access oxidative stress and try a potential therapeutic agent, vitamin C. In 2001, an important study that included 281 patients with documented coronary artery disease was published [115]. This study determined endothelium-dependent and -independent vasodilation by measuring forearm blood flow responses to acetylcholine and sodium nitroprusside. In addition, venous occlusion plethysmography and the effect of the coadministration of vitamin C (24 mg/min) were assessed in a subgroup of 179 patients. Besides confirming that the scavenging of increased oxygen-derived free radicals may account for the beneficial effects of vitamin C in CVD, an outstanding finding was that the positive response to vitamin C was an independent predictor of a higher risk of cardiovascular events. Heitzer et al. tested the effects of BH_4_ in chronic smokers in the presence and absence of vitamin C [116]. The study indicated that ascorbic acid could enhance endothelial ∙NO production in a BH_4_-dependent manner. It further suggested that the functional depletion of BH_4_ due to augmented BH_4_ oxidation rather than intracellular BH_4_ depletion contributes to endothelial dysfunction in chronic smokers [116]. Several studies showed that vitamin C improves endothelium-dependent vasodilation in patients with IDDM [117] and NIDDM [118,119,120].

### 7.2. Direct Measurements of the ROS and RNS

The production of ROS can be a relevant parameter for the determination of oxidative stress levels, although the direct detection of ROS is complicated because of their short lifetime. Electron paramagnetic resonance (EPR) is a spectroscopic method that can directly detect free radical species based on their paramagnetic properties [121,122,123]. Measurements can be performed in vitro, in vivo, or ex vivo. Unfortunately, the most precise measurement in vivo cannot be used for humans because the spin traps used for this determination are very toxic [121]. Another option is to collect the blood samples in the tubes with spin traps such as 5,5-dimethylpyrroline-N-oxide (DMPO) or spin probes such as 3-carboxy-2,2,5,5-tetramethylpyrrolin-1-hydroxide (CP-H), freeze them and measure them later [122,124]. Spin trapping is an almost “direct” method for free radical detection. [125] Spin trapping is based on the chemical reactions of the short-lived free radicals with the spin trap to form a more stable product with paramagnetic characteristics (spin adduct). Using the main characteristics of the EPR signal (e.g., doublet, triplet, and quartet pattern) and the hyperfine coupling profiles of the EPR signals, one can even determine the original nature of the trapped free radical (e.g., whether O_2_^∙−^ or hydroxyl radical). Most often, spin traps are nitrones or nitroso compounds, which are one electronic oxidation state higher than the nitroxides, and after reacting with the free radicals, they are converted into nitroxides. This direct measurement can also provide detailed mechanistic insights into the ROS reactions that involve free radical intermediates [121]. With the EPR method, a wide range of applications is available, such as an investigation of the cellular redox state, structural and dynamic properties of the biological membranes, oximetry, and many more [125]. Since this method allows direct measurements of ROS, critical players in DM complications, it can be used in the clinical setting as routine blood analysis as it would just require a simple blood withdrawal; this is, in most cases, included in the routine laboratory analysis of diabetic patients. Blood would need to be directly frozen in liquid nitrogen and proceed to the lab, where measurements would take 5–10 min, and the results would be available a few minutes later [126].

### 7.3. Measurements of the Oxidative Damage in Lipids, Proteins, and DNA Molecules

#### 7.3.1. Lipid Peroxidation

Lipid peroxidation is commonly used as a marker of lipids’ oxidative modification/damage. The primary way to investigate oxidative stress would be to measure the peroxidation rate of membrane lipids or fatty acids. Lipid peroxidation causes the breakdown of the lipids and yields multiple primary oxidation products, such as conjugated diens or lipid hydroperoxides, and secondary (fragmentation) products, including MDA, F2-isoprostane, 4-hydroxynonenal, exhaled pentane, ethane, and hexane [121,127].

Observing the diens is compelling because it reflects the cell membranes’ molecular reorganization by consuming the polyunsaturated fatty acids occurring during the primary phase of lipid peroxidation [128]. Moreover, lipid peroxides are also the markers of the initial free radicals’ reactions and specific markers for cellular membrane damage [121].

Other products are also used to determine lipid peroxidation but have the disadvantage of being secondary products. For instance, one of them, the MDA, is produced during fatty acid autoxidation. This substance is most often measured by its reaction with the thiobarbituric acid, which generates thiobarbituric acid reactive substances (TBARS). Even though the TBARS assay is not specific to MDA (this induces MDA overestimation), this method is used as a general marker for lipid peroxidation. Still, the results should be cautiously taken due to their nonspecificity. Therefore, it would be best to combine it with more specific methods such as O_2_^−^ HPLC measurements [121,129].

Many years ago, it was found that the F2 isoprostanes were formed during the peroxidation of arachidonic acid catalyzed by free radicals [128]. Researchers showed that quantifying the F2 isoprostane compounds could be suitable for endogenous lipid peroxidation and oxidant injury assessment [130,131].

The primary aim of determining plasma lipid peroxidation using MDA is to obtain insights into systemic oxidative stress. For example, Desco et al. found that higher levels of plasma MDA in diabetic patients can be diminished by xanthine oxidase inhibition through allopurinol treatment [132], indicating that elevation of MDA levels also reflects increased O_2_^∙−^ levels. The particular challenge here is sample collection and blood coagulation. So far, HPLC, LC-MS/MS, and GC-MS methods have proven to be specific and more sensitive for free and adducted MDA [133].

#### 7.3.2. Protein Modification

RONS induce modifications of proteins, e.g., the formation of the carbonyl groups in the amino acid side chains. An increase in carbonyl in the amino acid side chain is a hallmark of oxidative stress conditions [134]. Therefore, determining the carbonyl groups is a standard method for studying the oxidative post-translational protein modifications caused by the ROS/RNS (including the free radicals). Usually, the carbonyl/protein ratio is determined. This method is appealing because the half-life of the carbonyl groups in the amino acid side chain is longer when compared to the half-life of free radicals, thus opening a perspective for clinical use. In addition, the carbonyl group formation may accumulate over prolonged oxidative stress conditions, increasing the sensitivity.

One of the significant post-translational modifications in proteins that can easily be detected and serve as a marker for oxidative protein damage is the 3-nitrotyrosine (3-NT). 3-NT is produced following the reaction between the ONOO^−^ and RNS, such as the NO_2_ with the amino acid tyrosine [135,136].

Dimerization of amino acids (e.g., o-o′-dityrosine) represents another readout of protein oxidation. However, a disadvantage of this method may be that the reaction kinetics for dityrosine formation is not well studied. Hence, its quantification gives less precise insights into the in vivo concentrations of oxidants, thus not being as appealing for clinical use [121].

Thiols are organic sulfur derivatives recognized by their active sites’ sulfhydryl residues (-SH). The primary thiol substance in the body is the amino acid cysteine, which is easily oxidized. Oxidation of a cysteine residue can change a protein’s function. Thus, measuring the thiol status may be helpful as a mechanism-based biomarker. Biologically occurring thiols include low-molecular-weight thiols (cysteine and GSH) and protein thiols. The use of thiols as biomarkers is affected by the type of reactive species involved in the oxidative modification, the electrochemical environment (e.g., pH and cofactor availability), and the redox state, pKa value, and structural environment of the targeted thiol [137]. In addition to specific limitations for the thiols, which are only present intracellularly, the general problem is that the oxidation products of the protein cysteine residues are unstable, and thus measurement has to be conducted in a short period. Other thiols can quickly reduce them; therefore, it is necessary to treat them immediately with an alkylating agent to prevent further redox changes. Moreover, GSH level changes are not necessarily only due to oxidative stress but might also reflect a nutritional/metabolic imbalance, being a further disadvantage of this method. Plasma GSH levels could as well be affected by the GSH transporters, while cellular mechanisms, Nrf2, counteract the oxidative stress by increasing the GSH synthesis. Finally, GSSG concentrations are small and difficult to measure unless sensitive HPLC methods are used [138]. The latter could be of interest as a potential diagnostic tool, e.g., to measure the ratio of GSH and GSSG with an advanced HPLC method from patient plasma [139]. Already in 1989, scientists used this approach in patients. They applied a basic enzymatic assay where the activity of y-glutamylcysteine synthetase and glutathione synthetase was measured in the erythrocytes of T2DM patients. A decrease in the ratio of GSH/GSSG was found in these patients [140]. Measuring glutathione metabolites offers unique insights into the redox status in patients with mitochondrial dysfunction, including patients with DM [141].

#### 7.3.3. DNA Modification

The RONS induce different levels of DNA damage, such as strand breaks, DNA-protein cross-links, and base modifications. There are various methods for quantifying DNA modifications [142]. However, the most common marker for oxidative DNA damage is 8-hydroxy-2′-deoxyguanosine (8-OHdG), produced by the ROS/RNS (most efficiently by hydroxyl radicals) induced oxidation of the purine base in guanosine or its deoxy analog. 8-OHdG can be easily measured in blood or urine using commercial ELISA, HPLC, or LC-MS-based methods [121]. Since changes in epigenetic marks and differential regulation of epigenetic modulators have been observed in different models of diabetes and its associated complications, it is highly interesting to approach their roles in the pathogenesis of diabetes. Some essential epigenetic mechanisms that can alter gene expression are DNA methylation, histone post-translational modifications, and noncoding RNA-mediated pathways [143]. Extensive longitudinal studies with markers collected over time would greatly contribute to a better understanding of the role of epigenetic mechanisms in the evolution of a complex disease. Eventually, this could predict an individual’s risk and give access to targeted strategies to prevent and treat T2DM. Furthermore, the first steps are already made to identify potential DNA methylation biomarkers for T2DM. For example, decreased DNA methylation and enhanced gene expression of the Septin-9 (SEPT9) gene have been reported in pancreatic islets of T2D individuals [144]. In addition, scientists showed that DNA methylation changes in human blood cells and pancreatic islets are associated with insulin secretion [145]. Nevertheless, whether these tests can be used for DM needs to be further explored.

### 7.4. Antioxidants Measurement

#### 7.4.1. Enzymatic Antioxidant Activity

Quantifying the antioxidant activity of the SOD, catalase, and GPx is often used in preclinical and clinical studies. This method allows us to determine the antioxidant defense capacity under normal conditions and during oxidative stress [146]. The activity of antioxidants can be modified differently. It can be increased firstly (adaptation, counter-regulation) or decreased after prolonged oxidative stress (utilization, oxidative damage of antioxidant systems) [127]. This would be of high importance for the early detection of DM since abnormalities of chain-breaking antioxidant status exist in patients with DM before the development of detectable vascular disease [147]. This opens a new therapeutic perspective for preventing oxidative stress-induced organ damage.

#### 7.4.2. Antioxidant Vitamins

Determination of the non-enzymatic antioxidants, such as vitamins (vitamins A, C, and E) in plasma, is commonly used to assess the antioxidant capacity and vitamin deficiency [113,121]. Likewise, antioxidant enzymes and vitamin levels are altered during oxidative stress and can be used as markers [121]. Nonetheless, it must be acknowledged that vitamin C especially requires stabilizing additives in the samples to prevent autoxidation and pro-oxidative mechanisms [148]. In the case of vitamin E, which is transported in the blood in a form bound to lipids, the cholesterol/vitamin E ratio is often measured as a marker of vitamin E status [149]. In the clinical context, this technique is used to study the deleterious consequence of DM on the antioxidant status of non-enzymatic antioxidants such as vitamins A, C, and E. The activities of enzymatic and the levels of non-enzymatic antioxidants are usually significantly decreased in diabetic patients compared to healthy control subjects, as shown in [150].

#### 7.4.3. Other Antioxidants

Measuring the reduced and oxidized thiols in proteins is another technique for analyzing the antioxidant capacity and oxidative stress. In the same way, low molecular weight thiol compounds can also be determined. The method for measuring glutathione (GSH), which represents the most abundant thiol in the human body, and GSSG (oxidized form of GSH), is frequently used here to determine oxidative stress. This method is essential because the ROS/RNS oxidizes GSH into GSSG. Therefore, the ratio of GSH/GSSG is a commonly used parameter for assessing oxidative stress levels [151]. This method is also adopted in clinical use, and a significant decrease in GSH content in diabetic erythrocytes was observed. The decrease in GSH content represents increased utilization due to oxidative stress [150].

Uric acid also represents an essential oxidative stress marker [152]. However, it has some disadvantages since its concentration levels can vary because of oxidative stress, the purines cycle, and renal extractions. Therefore, uric acid alone is not a reliable oxidative stress marker. Nevertheless, allantoin, an oxidized product of uric acid, could be a useful endogenous marker of oxidative stress, considering that allantoin is absent from human body fluids under normal conditions. However, allantoin also has a disadvantage as an oxidative stress marker since it gets oxidized and degraded by the ROS/RNS in blood samples, which may cause an underestimation of the oxidative stress [153]. Therefore, studies suggested that the isocratic LC–MS method for plasma allantoin, which does not require extensive sample work-up or derivatization and has the advantages of speed, simplicity, and low running cost without compromising specificity, precision, or recovery, is advantageous. As a result, allantoin in patients with T2DM is clearly and markedly increased [154].

#### 7.4.4. Total Antioxidant Capacity

Due to the plethora of antioxidants in body fluids and tissues, it is not easy to measure one antioxidant separately. Therefore, some methods have been developed to measure the total antioxidant capacity (TAC) in biological samples [155]. The most commonly used method utilizes a pro-oxidant and measures how much of the ROS/RNS produced by the pro-oxidant can be trapped by the endogenous antioxidants in a given biological sample. [113,156] Likewise, stable free radicals with intense color are used, and the kinetics of the disappearance of the color upon the reaction of the stable free radical with endogenous antioxidants provide information on the TAC of the measured sample [157]. However, this method has limitations, as it is relatively unspecific since many factors can affect the TAC (e.g., also, considering whether the macromolecules are entirely removed from the sample or not, “deproteinization”), and it can also happen that changes in some of the antioxidants do not cause any TAC changes [158].

### 7.5. What Is the Best Method for ROS/RNS Detection?

Is there a perfect method to measure oxidative stress and detect ROS/RNS? Unfortunately, the answer to this question is not easy. Each method has its pros and cons. For example, a relatively nonspecific method may allow a high-throughput analysis to measure the large sample numbers at meager costs and without the need for sophisticated technical devices, such as a mass spectrometer or EPR spectroscope. In contrast, a particular method may have the disadvantage of being time-consuming, very expensive, and requiring technical devices not available in any laboratory. Moreover, the required sample volume or mass may represent a limitation for applying some techniques.

Furthermore, due to the complex nature of the experimental task, there is not only “one” measurement for all analytical problems in redox biology or oxidative stress research. Therefore, the combination of several assays may provide the best answer to a given analytical problem and allow adaptation to the available equipment and the approved budget. This is especially true for measuring antioxidative capacity or oxidative cell damage in patients suffering from DM. For example, scientists performed a study where blood samples from diabetic patients were collected by venous blood withdrawal in heparinized tubes, and the plasma was separated. Further, they performed the following:The biochemical investigation, including blood glucose, HbA1C, urea, creatinine, total protein, albumin, total cholesterol, triglyceride, HDL-C, and LDL-C, were determined;The determination of levels of serum thyroid stimulating hormone (TSH), total triiodothyroxine (T3), free thyroxine (FT4), and free triiodothyronine (FT3);Lipid peroxides were estimated by measurement of thiobarbituric acid reactive substances in plasma by the method of Yagi [159];SOD was assayed utilizing the technique of Kakkar et al. [160] based on inhibition of the formation of nicotine amide adenine dinucleotide, phenazine methosulfate, and amino blue tetrazolium formazan. CAT was assayed colorimetrically;GPx activity was measured by the method described by Rotruck et al. [161], and GSH was determined by the method of Ellman [162];Glutathione-S-transferase (GST) activity was determined spectrophotometrically by the method of Habig et al. [163], and plasma vitamin A (β-carotene) was estimated by the method of Bradley and Hombeck [164].

This approach indeed reflects an optimal way of approaching the early consequences of DM, but it would be too costly and time-consuming. Therefore, we suggest a more cost-effective and less time-consuming compromise where at least one previously described method for oxidative stress detection (one from antioxidants measurement, one protein modification, etc.) is used in combination with standard DM diagnostic methods.

Scientists recommend paying specific attention to the stability of the samples during the isolation and their preparation, isolation, storage, and analysis. This is a critical challenge when measuring oxidative modification since it can be influenced by exposure to ambient oxygen. Therefore, using more than one assay and including appropriate controls is crucial. Further, assays that directly and precisely measure the ROS of interest and its particular endpoints should be chosen before global oxidative stress measurements. Additionally, measured values of antioxidants (such as glutathione) or products of oxidation (such as lipid peroxidation products) should be presented as absolute values (in molar concentrations). This approach is crucial for inter-tissue comparisons and would increase the reproducibility of the assays [165].

Recently, an article with a summarized recommendation was published. Further 16 recommendations illustrated principles of cautious philosophy:

General recommendations:If possible, avoid the term ‘ROS’ and define the actual chemical species involved and their properties. If not, discuss caveats about the term ROS.Any putative effects of antioxidants should be chemically plausible and confirmed by measurements of oxidative damage.Selective generation of O_2_^−^ and H_2_O_2_ and specific inhibition/deletion of redox-active enzymes should be used to confirm the roles of these species.How oxidative damage to a biomolecule arises, is repaired, cleared, and is measured, should be discussed when presenting levels of oxidative damage markers.

Measurement of ROS:5.Use commercial kits only if the measured species and the detection method are explained and are chemically plausible.6.When using fluorescent ROS probes, the selectivity and potential artifacts should be clarified (especially for 2′,7′-dichlorodihydrofluorescein diacetate, DCFH-DA). In addition, controls should be carried out to show that the response is due to the proposed species, and orthogonal techniques should be used to corroborate the conclusion.7.ROS should not be ‘measured’ in tissue homogenates or cryosections.8.To detect O_2_
^−^ in vitro, use the SOD-sensitive reduction in cytochrome c, while aconitase inactivation within mitochondria can be used in vivo. Be cautious when using luminol or lucigenin.9.Hydroethidine or MitoSOX probes cannot be used to detect O_2_^∙−^ production by simple fluorescence measurements. Use specific identification of 2-HE products instead.10.Genetically encoded fluorescent probes are sensitive detectors of H_2_O_2_ in vivo. However, boronate probes are the preferred technique if their expression is impossible.

Measurement of oxidative damage:11.Application of the thiobarbituric acid-reactive substances (TBARS) assay to cells, tissues, or body fluids as the sole measure of lipid peroxidation is not recommended.12.Measuring F2-isoprostanes by LC-MS/MS with appropriate internal standards is the preferred biomarker of lipid peroxidation.13.Analysis of protein carbonyls by ELISA, fluorescein-5-thiosemicarbazide (FTC), and immunoblotting can detect 13 general oxidative protein damage. Orthogonal approaches to quantify individual oxidation products are encouraged.14.The oxidative modifications of nucleic acid can be measured using the comet assay on isolated cells and by UPLC-MS/MS for 80HdG and 80HG determination in body fluids or tissues.15.The use of antibodies to measure specific oxidation products must incorporate controls for nonspecific interactions and competitive data with authentic epitopes.16.Biomarkers must be measured to confirm that any antioxidants used decrease oxidative damage to the relevant biomolecules [166].

## 8. Discussion

Oxidative stress and impaired redox signaling are closely related to the initiation and progression of DM [127]. Therefore, in the clinical and diagnostic context, it is crucial to identify and quantify the ROS and RNS and to develop and validate the new methods for their tissue/cell/compartment-specific distribution and feasibility. However, despite all the mentioned facts, a reliable O_2_^−^ detection in biological samples per se and mainly in tissues or cellular compartments is still challenging [167]. The most evident problem with the commonly used probes is the low specificity and artificial production of the O_2_^∙−^signal by a mechanism involving the autoxidation and/or redox cycling caused by either the reactivity of the probe itself or their reactive intermediates with molecular oxygen [168].

This raises the question of why the specific O_2_^∙−^detection is essential. Adequate reaction to stressful conditions depends on the low formation rates of the O_2_^∙−^ and other ROS that provide a redox control of the essential cellular activity in a spatially and temporarily restricted manner [169]. In contrast, excessive and uncontrolled ROS formation causes dysfunction of the cells or even cell death, further leading to pathophysiological changes, organ damage, and the development of severe diseases [170]. Furthermore, many redox-sensitive structures are located in the mitochondrial matrix and intermembrane space at the mitochondrial level, such as iron–sulfur clusters, thiol-rich enzymes, and redox-regulated mitochondrial pores [171,172]. For that reason, the ROS and RNS in mitochondria can cause severe functional damage. Since different cellular mechanisms produce the ROS and RNS in separate cellular compartments, developing compartment-specific and species-specific probes for their detection seems essential. In this regard, there have been substantial advances in researching redox analytical assays in recent years, and several state-of-the-art assays are now available [173]. Figure 9 shows different products from the reaction of the O_2_^∙−^ with extracellular, intracellular, and mitochondrial dihydroethidium DHE analogs. These probes share the same chemistry regarding their reaction with the O_2_^∙−^. Kalyanaraman et al. anticipated in 2017 that if the DHE is attached to a peptide targeted to a nucleus, lysosome, or peroxisome, the O_2_^−^ and other oxidants formation in these organelles could be precisely determined [173]. Scientists also covalently attached the lysosome-targeting peptides to a boronate for measuring the specifically lysosomal H_2_O_2_ [174].

As mentioned previously, there may not be “one” optimal method for oxidative stress detection and quantification. Still, the presented data suggest that state-of-the-art HPLC-based O_2_^∙−^quantification methods nicely correlate with other well-accepted classical methods for ROS detection (e.g., dot blot and DHE staining) [175]. Thus, dihydroethidium (DHE)-HPLC, mitochondria-targeted fluorescence dye triphenylphosphonium-linked hydroethidium (mitoSox)-HPLC, and hydropropidine (HPr^+^)-HPLC-based O_2_^∙−^measurements represent the methods of choice for the O_2_^∙−^ detection in biological samples but correlate well with the more classical assays [129,175]. An essential advantage of the HPLC-based methods is the absolute quantification of the O_2_^∙−^ produced in different tissues or fluids. In addition, each method is specific for detecting the O_2_^∙−^ in separate cellular compartments, which can be of great importance for examining the exact mechanisms that cause oxidative stress and consequently allow for a particular treatment of oxidative stress-associated diseases. For DM, this is of specific importance because most of the O_2_^∙−^ originates from the mitochondria uncoupling; therefore, mitochondrial-specific O_2_^∙−^ detection would be a critical method for its early detection [175]. Further, it would be of great importance to couple an analytical method with an adequate one for assessing endothelial function. This approach would provide valuable insight into oxidative stress and the crucial adverse effects on the vasculature.

In the future, a significant improvement in the diagnostic of DM could be the establishment of an HPLC-based assay for the detection of the O_2_^∙−^ in different compartments within a given sample with only one analysis (e.g., incubation of tissue with DHE, mitoSOX, and HPr^+^ and detection of the superoxide-specific oxidation products within the same chromatogram). This method would allow specific superoxide targeting: This would consecutively reveal the exact role of oxidative stress in the pathogenesis of CVD observed in diabetic patients and the exact mechanism of how hyperglycemia induces CVD. Further, it would help to develop a new therapeutic strategy for DM and many other diseases that include oxidative stress, such as arterial hypertension, nitrate tolerance, cardiomyopathy, etc.

In summary, we suggest that early detection of oxidative stress would significantly contribute to standard routine methods of disease detection in DM. This would allow the detection of oxidative stress and associated inflammation at an early stage, thus preventing organ damage, as proposed in Figure 10. The respective new diagnostic tools would comprise specific techniques, such as new ROS detection methods (e.g., EPR and HPLC) [129], oxidative damage markers (MDA) [133], ROS sources (mitochondrial NOX2 expression) [175], and antioxidants defense system quantification (HPLC-GSH method) [139], which are now available.

Our approach would allow earlier detection of DM as well as diabetic cardiovascular sequelae and therefore contribute to developing better and more sophisticated therapeutic strategies concerning antioxidant therapies. Consequently, the onset of timely and more specific antioxidant therapy could further ameliorate the prognosis of DM.

Therefore, the aim of future research should involve the development of an HPLC-based methodology to allow an all-in-one detection: ROS, oxidative damage markers, NOX2 expression, and antioxidants. For further increase in the specificity of these redox biomarkers for early diabetes, their detection could be combined with a classical marker of hyperglycemia, e.g., components of AGE/RAGE signaling.

## Figures and Tables

**Figure 1 cells-12-00583-f001:**
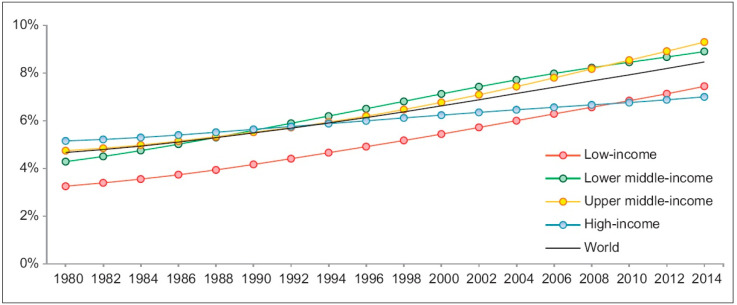
Trends in prevalence of DM from 1980 to 2014 by country income group. DM, diabetes mellitus [2] (permission number 391487).

**Figure 2 cells-12-00583-f002:**
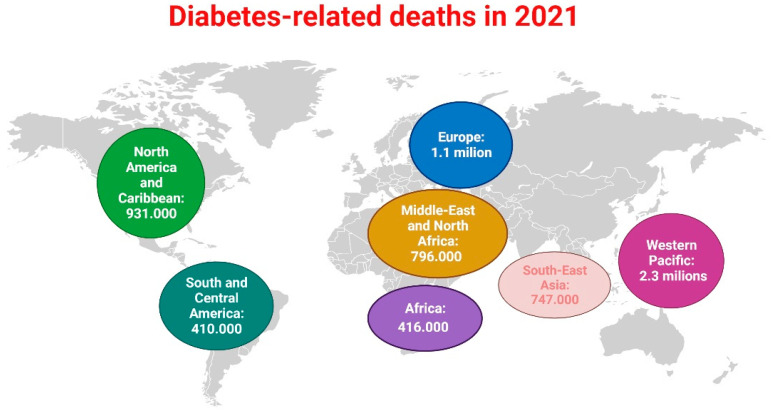
DM-related deaths (among 20–79 years old) in 2021. DM, diabetes mellitus. The figure is drawn with Biorender based on tabular data from reference [8].

**Figure 3 cells-12-00583-f003:**
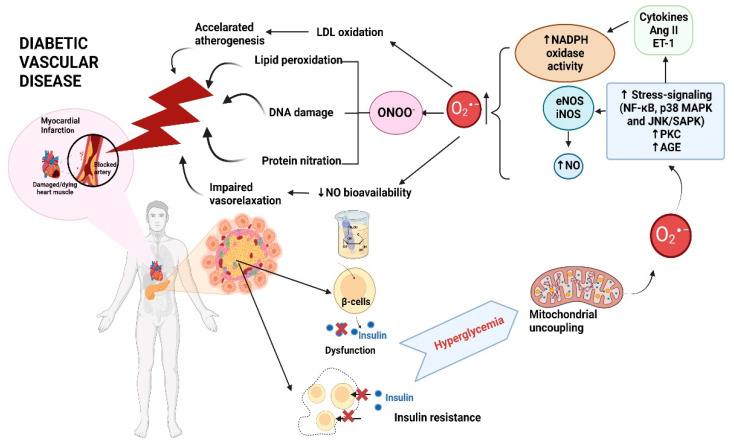
Schematic overview of purposed molecular mechanisms involved in DM and cardiovascular sequelae. AT II, angiotensin II; DM, diabetes mellitus; DNA, deoxyribonucleic acid; ET-1, endothelin 1; eNOS, endothelial nitric oxide synthase; iNOS, inducible nitric oxide synthase; LDL, low-density lipoprotein; NF-κB, nuclear factor kappa-light-chain-enhancer of activated B cells; ·NO, nitric oxide; O_2_^∙−^, superoxide anion radical; p38 MAPK, p38 mitogen-activated protein kinases; ONOO^−^, peroxynitrite; JNK/SAPK, stress-activated protein kinase/c-Jun NH(2)-terminal kinase. The figure is drawn with Biorender.

**Figure 4 cells-12-00583-f004:**
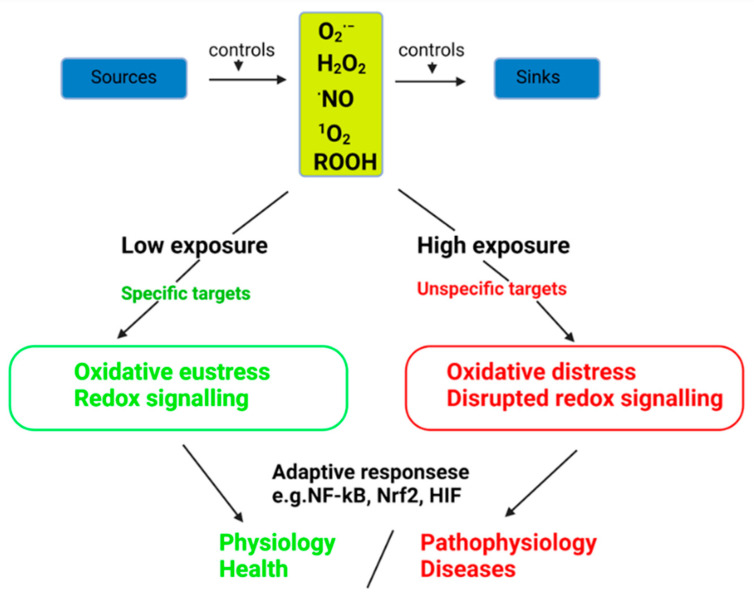
Oxidative stress and its relationship to redox signaling. Endogenous or exogenous sources produce different oxidants. The removal reactions (sinks) also control normal levels of oxidants. Low oxidants exposure induces redox changes in the specific targets leading to redox signaling (oxidative eustress), whereas high exposure leads to disturbed redox signaling and/or damage to biomolecules (oxidative distress). Adaptive responses modulate and counteract these pathways. Depending on the outcome, it contributes either to health or disease progression. O_2_^∙−^, superoxide anion radical; H_2_O_2_, hydrogen peroxide; ROOH, organic hydroperoxides; ∙NO, nitric oxide; ^1^O_2_, singlet molecular oxygen; Nrf2, nuclear factor erythroid 2–related factor 2; HIF, hypoxia-inducible factor. The figure is redrawn with Biorender from reference [26] with permission (License Number 5484750226313). Copyright © 2019 Elsevier Inc.

**Figure 5 cells-12-00583-f005:**
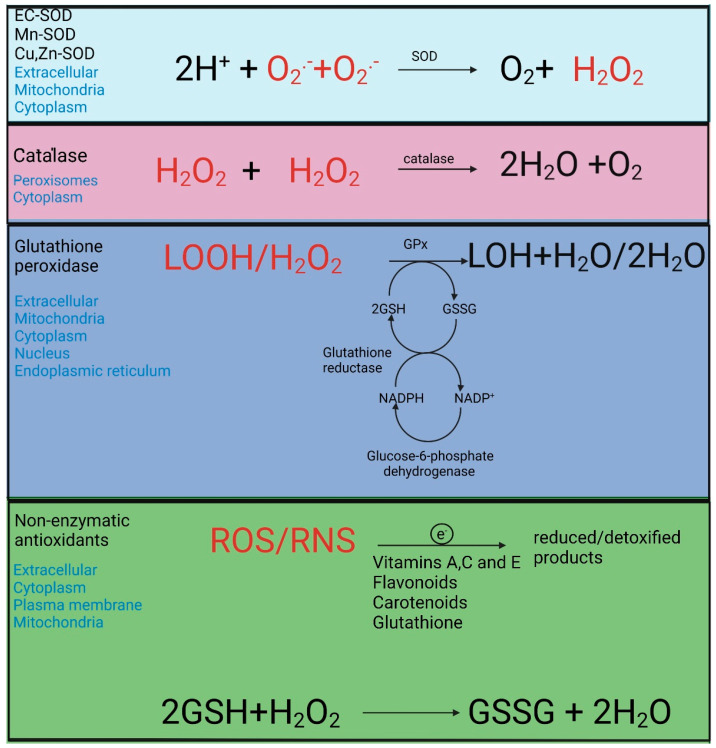
The antioxidant system in cells. ROS and RNS are neutralized by enzymatic and non-enzymatic antioxidants, which catalyze the degradation of these reactive species by transferring electrons to them (antioxidants are also named “electron donors”). The enzymatic antioxidants catalyze reactions that degrade the respective ROS or RNS. Thus, the superoxide dismutase (SOD) dismutases the O_2_^∙−^ to H_2_O_2_ and O_2_. Catalase and glutathione peroxidase (GPx) converts H_2_O_2_ to H_2_O (and oxygen in the case of catalase). The GPx also converts LOOH to lipid alcohols (LOH) and H_2_O_2_. The glutathione reductase (GR) restores GSH pools from oxidized glutathione (GSSG) using NADPH as reducing equivalents. The non-enzymatic antioxidants, such as vitamins, flavonoids, and glutathione, can also reduce and degrade ROS and RNS by transferring electrons to them. Cu, Copper; Cu, Zn-SOD, copper, zinc superoxide dismutase; EC, extracellular; EC-SOD- extracellular superoxide dismutase; GPx, glutathione peroxidase; GSH, glutathione; GSSG, glutathione disulfide; H_2_O_2_ hydrogen peroxide; H_2_O, water; LOH, lipid alcohols; LOOH, lipid hydroperoxides; Mn, manganese; Mn-SOD, manganese superoxide dismutase; NADPH, nicotinamide adenine dinucleotide phosphate; O_2_^∙−^, s superoxide anion radical; RNS, reactive nitrogen species ROS, reactive oxygen species. The figure is redrawn with Biorender from reference [39] with permission (License Number 5484750703800). Copyright © 2017, Springer International Publishing AG.

**Figure 6 cells-12-00583-f006:**
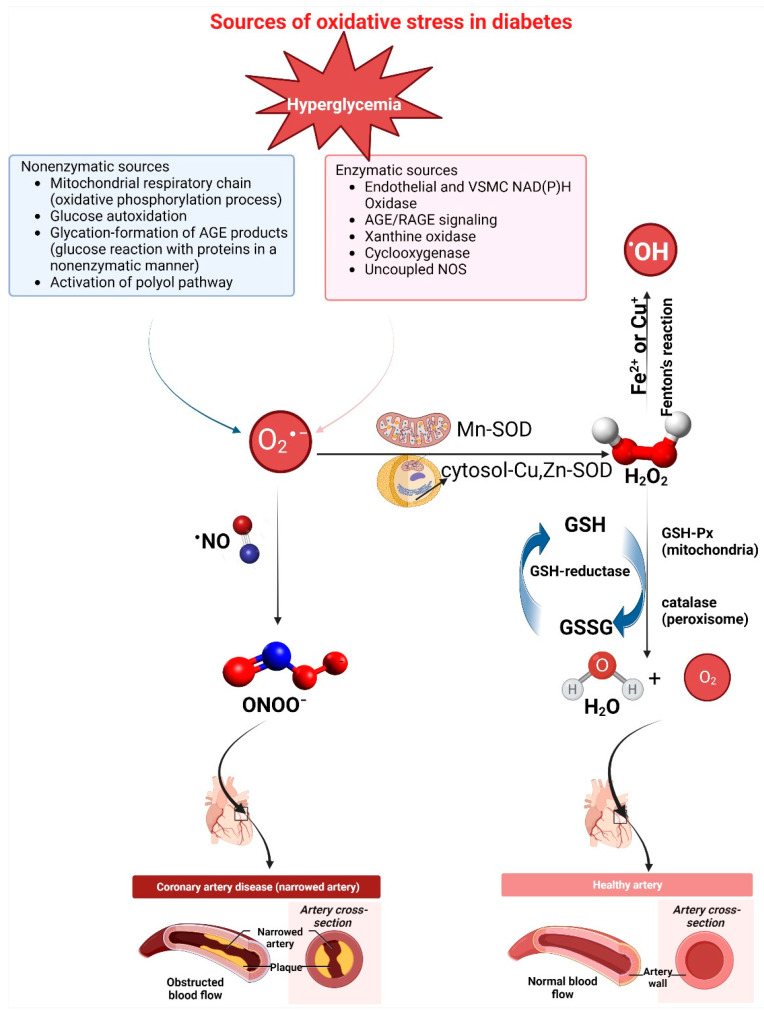
Generation of reactive species in DM. Some of the essential ROS and RNS are shown in the picture. Oxygen is converted to O_2_^∙−^ via the activation of enzymatic and non-enzymatic pathways, which is then dismutated to H_2_O_2_ by the superoxide dismutase (SOD). H_2_O_2_ can be converted to H_2_O by catalase or glutathione peroxidase (GSH-Px) or to ^∙^OH after a reaction with Cu or Fe. Glutathione reductase regenerates glutathione (GSH). In addition, O_2_^∙−^ reacts rapidly with ∙NO to form ONOO^−^. DM, diabetes mellitus; NO, nitric oxide; OH, hydroxyl radical; Cu, Copper; Cu-SOD, copper-superoxide dismutase; Fe, iron; GSH, glutathione; GSSG, glutathione disulfide; GSH-Px, glutathione peroxidase; Mn-SOD, manganese superoxide dismutase; O_2_, molecular oxygen; O_2_^∙−^, superoxide anion radical; H_2_O_2_, hydrogen peroxide; H_2_O, water molecule; ONOO^−^, peroxynitrite; VSMC, vascular smooth muscle cells. The figure is drawn with Biorender, based on data from reference [36] under the open access Creative Commons CC BY license.

**Figure 8 cells-12-00583-f008:**
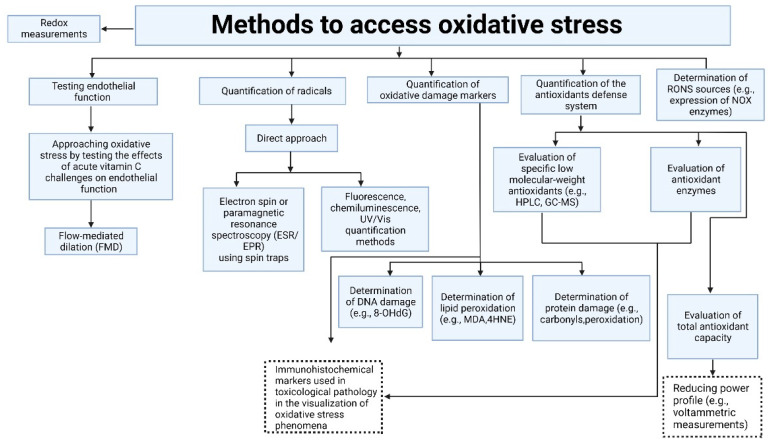
Methods to access oxidative stress. Schematic classification of the methods for determining and quantifying oxidative stress. 4HNE, 4-hydroxynonenal; FMD, flow-mediated dilation; GC-MS, gas chromatography-mass spectrometry; HPLC, high-performance liquid chromatography; MDA, malondialdehyde; NOX, NADPH oxidase; ROS, reactive oxygen species; RNS, reactive nitrogen species; UV, ultraviolet; VIS, visible; 8-OHdG, 8-hydroxydesoxyguanosin. Redrawn and modified from reference [112] with permission (License Number 5484770045925). Copyright © 2002, © SAGE Publications.

**Figure 9 cells-12-00583-f009:**
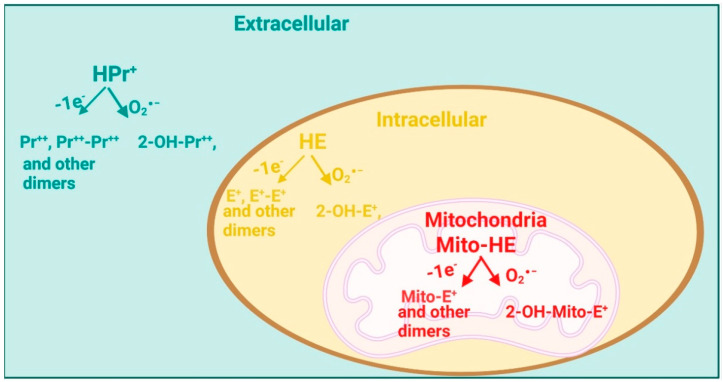
Detection of the site-specific formation of the O_2_^∙−^ and other oxidants using the DHE and its site-specific analogs. 2-OH-E^+^, 2-hydroxy-ethidium; 2-OH-Mito-E^+^, 2-hydroxy-mito-ethidium, 2-OH-Pr^++^, 2-hydroxy-propidium; E^+^, ethidium; E^+^-E^+^, ethidium dimer; HE, dihydroethidium; HPr^+^, hydropropidine; Mito-HE, mitoSox, triphenylphosphonium-linked hydroethidium; Pr^++^, propidium; Pr^++^-Pr^++^, propidium dimer. Redrawn and modified from reference [173] with permission (License Number 5484770645928). © 2016 Elsevier Inc.

**Figure 10 cells-12-00583-f010:**
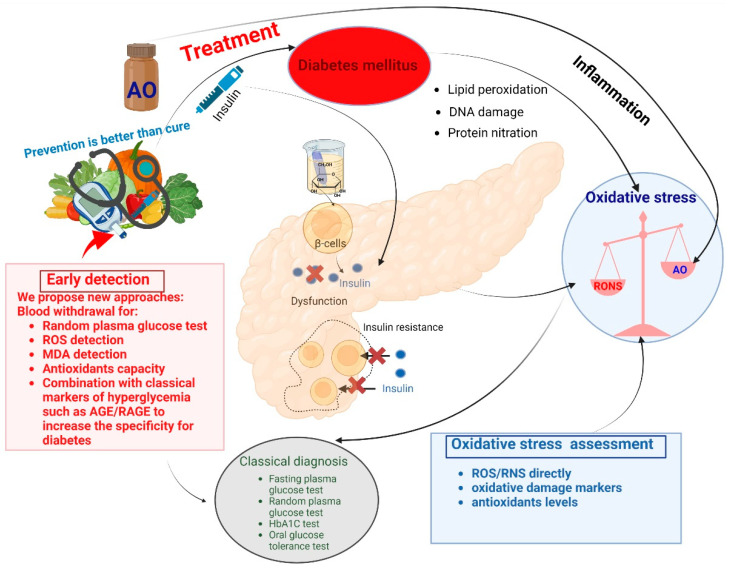
Schematic overview of the proposed methods. AO, antioxidants; DNA, deoxyribonucleic acid; MDA, malondialdehyde; RNS, reactive nitrogen species; RONS, reactive oxygen and nitrogen species; ROS, reactive oxygen species. The figure is drawn with Biorender.

## Data Availability

Review, no data available.

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
