# Peer review of "Early Detection Is the Best Prevention—Characterization of Oxidative Stress in Diabetes Mellitus and Its Consequences on the Cardiovascular System"

_cells, 2023, doi:10.3390/cells12040583_

Round 1

Reviewer 1 Report

The manuscript is a valuable and informative study revealing the relation between pathogenesis of cardiovascular disease (CVD) and diabetis mellitus especially DM2. Focusing on oxidative stress and the main sources of it, the study indicates the problems caused by inflammation and shows their importance for early diagnosis and treatment of the disease.  

There are numerous strengths to this study and I think the title is quite sufficient in terms of highlighting the points that the study is concentrated, and it indicates what you should focus on when you read the article.  

The manuscript will make an important contribution to the literature in terms of revealing a holistic approach to the development of diabetes caused by oxidative stress and cardiovascular diseases, which are one of its further consequences. It evaluates the topics mentioned in other published sources from general to specific and offers a specific perspective on the current situation of the subject.  

The introduction is relevant, sufficient and based on previous related research findings. Sufficient information about the previous study findings is presented. Also, the references are up-to-date and appropriate.

Author Response

Answers to Reviewer 1:

The manuscript is a valuable and informative study revealing the relation between pathogenesis of cardiovascular disease (CVD) and diabetis mellitus especially DM2. Focusing on oxidative stress and the main sources of it, the study indicates the problems caused by inflammation and shows their importance for early diagnosis and treatment of the disease. 

There are numerous strengths to this study and I think the title is quite sufficient in terms of highlighting the points that the study is concentrated, and it indicates what you should focus on when you read the article. 

The manuscript will make an important contribution to the literature in terms of revealing a holistic approach to the development of diabetes caused by oxidative stress and cardiovascular diseases, which are one of its further consequences. It evaluates the topics mentioned in other published sources from general to specific and offers a specific perspective on the current situation of the subject. 

The introduction is relevant, sufficient and based on previous related research findings. Sufficient information about the previous study findings is presented. Also, the references are up-to-date and appropriate.

We are very grateful for the reviewer's recognition of our paper and positive feedback, and we hope that with additional changes we were able to further improve our manuscript (see requested changes of Revievers #2 and 3).

Reviewer 2 Report

Reviewer comments

Journal name: Cells-MDPI

Article title    : Early detection is the best prevention- Characterization of oxidative stress in diabetes mellitus and its consequences on the cardiovascular systemManuscript type: Review

General Comments:

In this review, the authors demonstrated the important role of oxidative stress in the pathogenesis of the cardiovascular disease (CVD) in diabetic patients as a consequence of hyperglycaemia. They have given an updated information about CVD , the remain leading cause of premature death in the western world and correlated the diabetes mellitus-associated oxidative stress and subsequent inflammation should be recognized at the earliest possible  stage to start with the appropriate treatment before the onset of the cardiovascular sequelae such as arterial hypertension or coronary artery disease (CAD). The authors have also given a clear-cut idea on the pathophysiology comprises increased reactive oxygen and nitrogen species (RONS) production by enzymatic sources and non-enzymatic sources, e.g. mitochondria, an uncoupled nitric oxide synthase, xanthine oxidase, and the NADPH oxidase. Considering that RONS originate from different cellular mechanisms in separate cellular compartments, adequate, sensitive, and compartment-specific methods for their quantification are crucial for early detection towards the treatment of diabetes.

The authors have elaboratively given sufficient information mechanistically with proper diagrammatic representation, very interesting and understandable to the readers. Overall, the review work is of quality and presented clearly and worth to be considering for publication in Cells.

However,  the paper needs minor revision before to consider in this journal.

1.     There are minor corrections that need to be carried out throughout the manuscript. For instance, page no. 13, line 5, “in vivo and ex vivo” should be in italics.

2.     References are not as per journal format and it should be corrected accordingly. For example Journal name should be formatted throughout the references.

3.     Please provide a good resolution image for “Fig.5” and font size accordingly.

4.     Page no. 12, lines 391-395 should be aligned properly and check once.

5.     Page no, 17 lines 647-650 needs to be corrected.

6.     Minor spell check throughout the manuscript needs to be done. 

*********

Author Response

Answers to Reviewer 2:

General Comments:

In this review, the authors demonstrated the important role of oxidative stress in the pathogenesis of the cardiovascular disease (CVD) in diabetic patients as a consequence of hyperglycaemia. They have given an updated information about CVD , the remain leading cause of premature death in the western world and correlated the diabetes mellitus-associated oxidative stress and subsequent inflammation should be recognized at the earliest possible  stage to start with the appropriate treatment before the onset of the cardiovascular sequelae such as arterial hypertension or coronary artery disease (CAD). The authors have also given a clear-cut idea on the pathophysiology comprises increased reactive oxygen and nitrogen species (RONS) production by enzymatic sources and non-enzymatic sources, e.g. mitochondria, an uncoupled nitric oxide synthase, xanthine oxidase, and the NADPH oxidase. Considering that RONS originate from different cellular mechanisms in separate cellular compartments, adequate, sensitive, and compartment-specific methods for their quantification are crucial for early detection towards the treatment of diabetes.

The authors have elaboratively given sufficient information mechanistically with proper diagrammatic representation, very interesting and understandable to the readers. Overall, the review work is of quality and presented clearly and worth to be considering for publication in Cells.

However, the paper needs minor revision before to consider in this journal.

  1. There are minor corrections that need to be carried out throughout the manuscript. For instance, page no. 13, line 5, “in vivo and ex vivo” should be in italics.

We are thankful to the reviewer for this comment and put “in vivo” and “ex vivo” in italics throughout the manuscript.

  1. References are not as per journal format and it should be corrected accordingly. For example Journal name should be formatted throughout the references.

We changed the referenced to the journals format.

  1. Please provide a good resolution image for “Fig.5” and font size accordingly.

We provided a good resolution image for Fig.5 and changed font size, accordingly.

  1. Page no. 12, lines 391-395 should be aligned properly and check once.

Thank you for this suggestion. We aligned these lines and checked them.

  1. Page no, 17 lines 647-650 needs to be corrected.

We corrected the mentioned lines. Thank you.

  1. Minor spell check throughout the manuscript needs to be done.

We did a complete spell check of the manuscript.

Reviewer 3 Report

The review article written by, Sanela et al on “Early detection is the best prevention- characterization of oxidative stress in diabetes mellitus and its consequences on the cardiovascular system”. In short, the aim of the study is clear, however, the reviewer has concern about the significant outcome of the study.

The reviewer have few suggestions that could improve the manuscript.

1)    The manuscript title is more informative, However, as the text talks about in a different.

The main concept of this review which is shown on Fig 10. But, discussed by the authors in the review about early detection methods are very old and it’s not only relevant to DM and its complications of CVD.

2)    The authors discuss in detail literatures survey about major CVD markers which are linked with RNS/RONS.

3)    Fig 1 includes the details until current year for trends in prevalence of DM 1980-2022.

Overall, all the content in this review topic is very dense, but it is less informative.  The information in the text includes recent advancement in the field is needed to discuss in more detail.

Author Response

Answers to Reviewer 3:

The review article written by, S. Rajlic et al. on "Early detection is the best prevention- characterization of oxidative stress in diabetes mellitus and its consequences on the cardiovascular system". In short, the aim of the study is clear, however, the reviewer has concern about the significant outcome of the study.

  1. The manuscript title is more informative, However, as the text talks about in a different.

The main concept of this review which is shown on Fig 10. But, discussed by the authors in the review about early detection methods are very old and it's not only relevant to DM and its complications of CVD.

We are thankful to the reviewer for this comment. Indeed, this review's title is "catchy" in order to attract the attention of the readers. Furthermore, we are convinced that the title properly reflects the key messages of the manuscript, as we report and discuss the newest methods with respect to the characterization of oxidative stress, a key component of diabetes mellitus development and associated disease (e.g. CVD). We fully agree that a large part of diabetic pathophysiology is related to cardiovascular sequelae (reflected by our Figure 3 which summarizes the oxidative stress-initiated diabetic and cardiovascular complications), which without any doubt contribute to oxidative stress and accordingly show overlap with the redox markers for diabetes itself. We made this clearer in the legend of Figure 3 (page: 3, line: 77) by adding “cardiovascular sequelae”. However, we think that initial formation of reactive oxygen and nitrogen species comes from hyperglycemia-related processes (Figure 6) that cause further damage to the islet cells, thereby aggravating the diabetic condition with a further increase of oxidative stress. Only later, secondary cardiovascular damage by development of hypertension and progression of atherosclerosis will contribute to oxidative stress conditions making it impossible to distinguish between the different causalities. This important notion was now added to the end of section 1 (pages: 3-4, lines 96-109): “Of note, large part of diabetic pathophysiology is related to cardiovascular sequelae, which without any doubt contribute to oxidative stress and accordingly show overlap with the redox markers for diabetes itself. However, we think that the initial formation of reactive oxygen and nitrogen species comes from hyperglycemia-related processes that cause further damage to the islet cells, thereby aggravating the diabetic condition with a further increase of oxidative stress. Only later, secondary cardiovascular damage by development of hypertension and progression of atherosclerosis will contribute to oxidative stress conditions making it impossible to distinguish between the different causalities. Therefore, combination of the measurement of oxidative stress markers or measurement of ROS (reactive oxygen species) and harmful RNS (reactive nitrogen species) with classical markers of hyperglycemia-driven pro-oxidative pathways (e.g., advanced glycation end products (AGE) and their receptor (RAGE)) may help to more specifically identify and correctly assign oxidative stress markers for early diabetes.” The focus of our review is on new methods to detect oxidative stress, not detection of diabetes mellitus itself. Therefore, in Figure No.10 we summarize these techniques and suggest a new diagnostic approach, offering the possibilities to start prevention strategies at a very early stage of  disease development.

However, in order to address the reviewers concern, we have added information on the new diagnostic methods of diabetes mellitus and associated organ or tissue dysfunction (e.g. nerves) in the introduction (page: 3, lines: 83-92): “The current diagnosis of prediabetes and diabetes is based on blood glucose or hemoglobin A1c (HbA1c) test. However, both methods are invasive and time-consuming, with high costs. Also, methylglyoxal (MG) is evolving as a DM marker because it plays a vital role in biological processes. In addition to glucose in human blood, there are also considerable amounts in other biofluids such as saliva, tears, sweat, and interstitial fluids (ISF), but scientists showed that ISF reflects the change of blood glucose level with an unavoidable delay. Furthermore, since it was shown that early damage of small nerves could occur during the progression of T2DM, some devices have emerged to assess small-fiber autonomic dysfunction . On top of the mentioned approaches, others are mentioned in the literature, but none found a place in the clinical use.”

  1. The authors discuss in detail literatures survey about major CVD markers which are linked with RNS/RONS.

We are thankful to the reviewer for pointing this out. We did use this approach in our complete manuscript because we support the idea that this will allow earlier detection of diabetes mellitus as well as diabetic cardiovascular sequelae and therefore contributes to the development of better and more sophisticated therapeutic strategies concerning antioxidant therapies. In order to make this clearer to the readers, we included the following sentence in the discussion (page: 19, lines: 773-775). Our approach would allow earlier detection of DM as well as diabetic cardiovascular sequelae and therefore contribute to developing better and more sophisticated therapeutic strategies concerning antioxidative therapies."

As said above and added to the MS, a combination of oxidative stress markers with classical hyperglycemia markers such as AGE/RAGE components may be the straightforward approach for early and specific diagnosis of diabetes. This important point was now also added at the end of the discussion (page: 19, lines: 780-782): “For further increase of the specificity of these redox biomarkers for early diabetes, their detection could be combined with a classical marker of hyperglycemia, e.g. components of AGE/RAGE signaling. “

We also added the following text to the text box named “Early detection” on Figure 10, left lower corner (page: 20): “Combination with classical markers of hyperglycemia such as AGE/RAGE to increase the specificity for diabetes."

  1. Fig 1 includes the details until current year for trends in prevalence of DM 1980-2022. Overall, all the content in this review topic is very dense, but it is less informative. The information in the text includes recent advancement in the field is needed to discuss in more detail.

We are grateful for this comment. Figure 1 shows the prevalence from 1980-2014. This is the latest WHO-published report and therefore the most recent data with respect to prevalence. Further, the WHO is the most recognizable institution worldwide for this type of information. We received permission to publish this graph, which we included in our manuscript to emphasize the clinical and socio-economic importance of our review.

Furthermore, we discussed the newest methods of oxidative stress detection against the background of disease development and inflammation associated with diabetes mellitus in detail. We even summarized this reflection in Figure 8. Detailed information of underlying molecular and cellular mechanisms are provided in Figures 5-7. Therefore, we not only provided this information in the text of the manuscript but also took a great effort to interpret this data graphically and thus make it easily visible to a broad readership.

Reviewer 4 Report

·        Check the entire manuscript and correct any grammatical errors.

·        Below are some basic minor suggestions:

Line no. 18: Give a hyphen for the word non-enzymatic

Line no. 19: Mention the full form of NADPH

Line no. 30: Give a hyphen for the word insulin sensitive

Line no.39: Remove space before full stop after reference

Line no. 48, and 49: Avoid repeated words like “by”

Line no. 60: Avoid “and” several times

Line no. 62: Check the word spell “analogues”

Line no. 81: Add an abbreviation the very first time then continue it throughout the manuscript

Line no. 82: Add the abbreviation for ROS

Line no. 115: Mention the reference after the sentence ends

Line no. 122: Remove space before full stop after reference

Line no. 124: Mention the reference after the sentence ends

Line no. 131: Avoid the repeated word “to”

Line no. 149: Remove the word “therefore”

Line no. 154: Mention the abbreviation at line no. 19 for NADPH then continue it throughout the manuscript

Line no. 177: Check the starting of the word using “Of note”

Line no. 185: Mention the full form of FADH2

Line no. 186: Mention the full form of the “ATP”

Line no. 198: Check the spell “heme”

Line no. 212: Mention the reference after the sentence ends

Line no. 285: re-write the line

Line no. 291: Mention the reference after the sentence ends

Line no. 328: replace was as a were

Line no. 336: Add a reference

Line no. 388: Mention the full form of VCAM

Line no. 391, 393, 394, 396: Start the sentence with a capital word

Line no. 403: Just mention reference only

Line no. 404, 405, 406: Mention the abbreviation in the bracket

Line no. 410: Mention the full form of CVDs

Line no. 415: Change the word “interesting”

Line no. 429: Mention the full form of IDDM and NIDDM

Line no. 504: Check the word “adduct”

Line no. 510: Mention the reference after the sentence ends

Line no. 536: Remove two full stops

Line no. 558: Remove the word “actually”

Line no. 578: Add reference before full stop

Line no. 648, 650, 657: Start the sentence with a capital letter

Author Response

Answers to Reviewer 4:

Check the entire manuscript and correct any grammatical errors.

We agree with the reviewer and also found some misspellings and semantic errors. The manuscript was therefore subjected to a native 'speakers' correction.

  1. Line no. 18: Give a hyphen for the word non-enzymatic.

We are very thankful for pointing out this mistake and apologize for missing this in the first place. We corrected this mistake accordingly (page: 1, lines: 18-19).

  1. Line no. 19: Mention the full form of NADPH

We provided the full term of NADPH (page: 1, line: 20).

  1. Line no. 30: Give a hyphen for the word insulin sensitive

Thank you, we changed this as requested (page: 1, line: 31).

  1. Line no.39: Remove space before full stop after reference

We removed the space before the full stop after the respective reference (page: 1, line: 40).

  1. Line no. 48, and 49: Avoid repeated words like "by"

Thank you for this comment. We simplified the following sentence “The total number of deaths combined the number of annual all cause deaths due to DM related disease in each geographic region stratified by age and sex as well as age- and sex-specific mortality relative risks in people with and without DM. “ (page: 2, lines: 49-52).

  1. Line no. 60: Avoid "and" several times

This is right, thank you. We changed the mentioned passage as suggested (page: 2, lines: 63-64).

  1. Line no. 62: Check the word spell "analogues"

We corrected the misspelled word (page: 3, line: 66).

  1. Line no. 81: Add an abbreviation the very first time then continue it throughout the manuscript

We are very thankful for this comment and corrected our mistake (page: 3, line: 96).

  1. Line no. 82: Add the abbreviation for ROS

In the corrected version, we added a paragraph at the end of the preceding section, and there the abbreviation for ROS is already included (page: 4, lines: 105-106).

  1. Line no. 115: Mention the reference after the sentence ends

We did as requested. Thank you (page: 5, line: 146).

  1. Line no. 122: Remove space before full stop after reference

Thank you. We removed the space before the full stop after the reference (page: 5, line: 150).

  1. Line no. 124: Mention the reference after the sentence ends

We mentioned the reference after the end of the sentence (page: 5, line: 153).

  1. Line no. 131: Avoid the repeated word "to"

Thank you, we corrected this (page: 5, lines: 159-161).

  1. Line no. 149: Remove the word "therefore"

We removed the word “therefore.” Thank you (page: 6, line: 177).

  1. Line no. 154: Mention the abbreviation at line no. 19 for NADPH then continue it throughout the manuscript

We mentioned the abbreviation “NADPH” in the respective line and further used this abbreviation within the whole manuscript (page: 1, line: 20).

  1. Line no. 177: Check the starting of the word using "Of note"

We removed the word "of note" (page: 7, line: 205).

  1. Line no. 185: Mention the full form of FADH2

We mentioned the full name of FADH2, as requested (page: 7, line: 213).

  1. Line no. 186: Mention the full form of the "ATP"

We mentioned the full name of ATP (page: 7, lines: 214-215).

  1. Line no. 198: Check the spell "heme"

We changed the word "heme" by "haem" (page: 7, line: 227).

  1. Line no. 212: Mention the reference after the sentence ends

The reviewer is correct. We now mention the reference after the sentence end (page: 8, line: 252).

  1. Line no. 285: rewrite the line

We rewrote the following sentence: "This observation was supported by in vitro studies which showed changes in PKC-specific activities as well as in its isoforms and DAG levels in diabetic rats." (page: 10, lines: 315-317).

  1. Line no. 291: Mention the reference after the sentence ends

We are very thankful for pointing out this mistake. We have corrected it (page: 10, line: 324).

  1. Line no. 328: replace was as a were

We replaced the word “was” with “were” (page: 11, line: 361).

  1. Line no. 336: Add a reference

We added reference No.96 (page: 11, line: 363).

  1. Line no. 388: Mention the full form of VCAM

We mentioned the abbreviation “VCAM” already before (page: 8, line: 255).

  1. Line no. 391, 393, 394, 396: Start the sentence with a capital word

We introduced the capital letter at the start of the sentence (page: 12, lines: 424, 426, 427, 429).

  1. Line no. 403: Just mention reference only

We are very thankful for this observation. Therefore, we indicated only [Reference No.] in the legends of figures that were adopted, but wrote “modified” from [Reference No] when the figure was modified.

  1. Line no. 404, 405, 406: Mention the abbreviation in the bracket

We are very thankful for this comment. This is a figure legend and we separated abbreviations from the full explanation with commas. We consistently used this format for each figure legend.

  1. Line no. 410: Mention the full form of CVDs

This abbreviation was already mentioned in abstract (page: 1, line: 12).

  1. Line no. 415: Change the word "interesting

We replaced the word “interesting” by “important” (page: 13, lines: 449-450).

  1. Line no. 429: Mention the full form of IDDM and NIDDM

For both abbreviations, the full names were previously mentioned in the text: IDDM (page: 11, lines:360-361) and NIDDM (page 11, line: 385).

  1. Line no. 504: Check the word “adduct”

We replaced the word “adduct” by “post-translational modification in proteins” (page: 14, line: 538).

  1. Line no. 510: Mention the reference after the sentence ends

We moved the reference to the end of the sentence (page: 14, line: 541). Thank you.

  1. Line no. 536: Remove two full stops

We removed two full stops as requested (page: 15, line: 571).

  1. Line no. 558: Remove the word "actually"

We removed the word “actually” (page: 15, line: 593).

  1. Line no. 578: Add reference before full stop

We moved the reference to the end of the sentence (page: 15, line: 604).

  1. Line no. 648, 650, 657: Start the sentence with a capital letter

Thank you. We started the sentences with a capital letter (page: 17, lines:683, 685, 692).

Round 2

Reviewer 3 Report

The revised manuscript improved better. However, a few minor corrections below will help to improve the quality of the manuscript.

1.     The authors need to use recent references and discussed in “best method for ROS/RNS detection”.  Few examples are., 

Griendling KK, Touyz RM, Zweier JL, Dikalov S, Chilian W, Chen YR, Harrison DG, Bhatnagar A; American Heart Association Council on Basic Cardiovascular Sciences. Measurement of Reactive Oxygen Species, Reactive Nitrogen Species, and Redox-Dependent Signaling in the Cardiovascular System: A Scientific Statement From the American Heart Association. Circ Res. 2016 Aug 19;119(5):e39-75. doi: 10.1161/RES.0000000000000110. Epub 2016 Jul 14. PMID: 27418630; PMCID: PMC5446086.

Murphy, M.P., Bayir, H., Belousov, V. et al. Guidelines for measuring reactive oxygen species and oxidative damage in cells and in vivo. Nat Metab 4, 651–662 (2022). https://doi.org/10.1038/s42255-022-00591-z

2.     Line 781, the author mentioned “proposed advanced new methods”, change to proposed methods.

Author Response

Answers to Reviewer 3:

The revised manuscript improved better. However, a few minor corrections below will help to improve the quality of the manuscript.

  1. The authors need to use recent references and discussed in “best method for ROS/RNS detection”. Few examples are.,

Griendling KK, Touyz RM, Zweier JL, Dikalov S, Chilian W, Chen YR, Harrison DG, Bhatnagar A; American Heart Association Council on Basic Cardiovascular Sciences. Measurement of Reactive Oxygen Species, Reactive Nitrogen Species, and Redox-Dependent Signaling in the Cardiovascular System: A Scientific Statement From the American Heart Association. Circ Res. 2016 Aug 19;119(5):e39-75. doi: 10.1161/RES.0000000000000110. Epub 2016 Jul 14. PMID: 27418630; PMCID: PMC5446086.

Murphy, M.P., Bayir, H., Belousov, V. et al. Guidelines for measuring reactive oxygen species and oxidative damage in cells and in vivo. Nat Metab 4, 651–662 (2022). https://doi.org/10.1038/s42255-022-00591-z

We are thankful to the reviewer for these additional suggestions. We have included the recommendations from Griendling KK et al.: (page: 17, lines 702-712): “Scientists recommend paying specific attention to the stability of the samples during the isolation and their preparation, isolation, storage and analysis. This is a critical challenge when measuring oxidative modification since it can be influenced by exposure to ambient oxygen. Therefore, using more than one assay and including appropriate controls is crucial. Further, assays that directly and precisely measure the ROS of interest and its particular endpoints should be chosen before global oxidative stress measurements. Additionally, measured values of antioxidants (such as glutathione) or products of oxidation (such as lipid peroxidation products) should be presented as absolute values (in molar concentrations). This approach is crucial for inter-tissue comparisons and would increase the reproducibility of the assays.”

Further we also showed an excellent summary of the 16 recommended principles by Murphy MP et al. (page: 18, lines: 713-760): “Recently an article with a summarized recommendation was published. Further 16 recommendations illustrated principles of cautious philosophy:

General recommendations:

  1. If possible, avoid the term 'ROS' and define the actual chemical species involved and their properties. If not, discuss caveats about the term ROS.
  2. Any putative effects of antioxidants should be chemically plausible and confirmed by measurements of oxidative damage.
  3. Selective generation of O2∙− and H₂O2, and specific inhibition/deletion of redox-active enzymes, should be used to confirm the roles of these species.
  4. How oxidative damage to a biomolecule arises, is repaired and cleared and is measured should be discussed when presenting levels of oxidative damage markers.

Measurement of ROS:

  1. Use commercial kits only if the measured species and the detection method are explained and are chemically plausible.
  2. When using fluorescent ROS probes, the selectivity and potential artifacts should be clarified (especially for 2',7'-dichlorodihydrofluorescein diacetate, DCFH-DA). In addition, controls should be done to show that the response is due to the proposed species, and orthogonal techniques should be used to corroborate the conclusion.
  3. ROS should not be 'measured' in tissue homogenates or cryosections.
  4. To detect O2∙− in vitro, use the SOD-sensitive reduction of cytochrome c, while aconitase inactivation within mitochondria can be used in vivo. Be cautious when using luminol or lucigenin.
  5. Hydroethidine or MitoSOX probes cannot be used to detect O2∙− production by simple fluorescence measurements. Use specific identification of 2-HE products instead.
  6. Genetically encoded fluorescent probes are sensitive detectors of H₂O2 in vivo. However, boronate probes are the preferred technique if their expression is impossible.

Measurement of oxidative damage:

  • Application of the thiobarbituric acid-reactive substances (TBARS) assay to cells, tissues or body fluids as the sole measure of lipid peroxidation is not recommended.
  • Measuring F2-isoprostanes by LC-MS/MS with appropriate internal standards is the preferred biomarker of lipid peroxidation.
  1. Analysis of protein carbonyls by ELISA, fluorescein-5-thiosemicarbazide (FTC) and immunoblotting can detect 13 general oxidative protein damage. Orthogonal approaches to quantify individual oxidation products are encouraged.
  2. The oxidative modifications of nucleic acid can be measured using the comet assay on isolated cells and by UPLC-MS/MS for 80HdG and 80HG determination in body fluids or tissues.
  • The use of antibodies to measure specific oxidation products must incorporate controls for non-specific interactions and competitive data with authentic epitopes.
  • Biomarkers must be measured to confirm that any antioxidants used decrease oxidative damage to the relevant biomolecules.”

  1. Line 781, the author mentioned “proposed advanced new methods”, change to proposed methods.

We agree with the reviewer. We changed to the “proposed methods” (page: 21, line: 847): Schematic overview of the proposed methods.
